# *Atopobium vaginae* and *Prevotella bivia* Are Able to Incorporate and Influence Gene Expression in a Pre-Formed *Gardnerella vaginalis* Biofilm

**DOI:** 10.3390/pathogens10020247

**Published:** 2021-02-20

**Authors:** Joana Castro, Aliona S. Rosca, Christina A. Muzny, Nuno Cerca

**Affiliations:** 1Centre of Biological Engineering (CEB), Laboratory of Research in Biofilms Rosário Oliveira (LIBRO), Campus de Gualtar, University of Minho, 4710-057 Braga, Portugal; joanacastro@ceb.uminho.pt (J.C.); aliona.rosca@ceb.uminho.pt (A.S.R.); 2Division of Infectious Diseases, University of Alabama at Birmingham, Birmingham, AL 35294, USA; cmuzny@uabmc.edu

**Keywords:** bacterial vaginosis, *Gardnerella* spp., *Atopobium vaginae*, *Prevotella bivia*, polymicrobial biofilms, virulence

## Abstract

Bacterial vaginosis (BV) is associated with a highly structured polymicrobial biofilm on the vaginal epithelium where *Gardnerella* species presumably play a pivotal role. *Gardnerella vaginalis*, *Atopobium vaginae*, and *Prevotella bivia* are vaginal pathogens detected during the early stages of incident BV. Herein, we aimed to analyze the impact of *A. vaginae* and *P. bivia* on a pre-established *G. vaginalis* biofilm using a novel in vitro triple-species biofilm model. Total biofilm biomass was determined by the crystal violet method. We also discriminated the bacterial populations in the biofilm and in its planktonic fraction by using PNA FISH. We further analyzed the influence of *A. vaginae* and *P. bivia* on the expression of key virulence genes of *G. vaginalis* by quantitative PCR. In our tested conditions, *A. vaginae* and *P. bivia* were able to incorporate into pre-established *G. vaginalis* biofilms but did not induce an increase in total biofilm biomass, when compared with 48-h *G. vaginalis* biofilms. However, they were able to significantly influence the expression of *HMPREF0424_0821*, a gene suggested to be associated with biofilm maintenance in *G. vaginalis*. This study suggests that microbial relationships between co-infecting bacteria can deeply affect the *G. vaginalis* biofilm, a crucial marker of BV.

## 1. Introduction

Bacterial vaginosis (BV) is the most-common vaginal infection affecting fertile, premenopausal, and pregnant women [1]. It is associated with important adverse outcomes related to pregnancy [2] and infertility [3]. Additionally, it is associated with an increased risk of acquisition of HIV and other sexually transmitted infections (STIs) [4,5,6,7,8]. Microbiologically, BV is a complex polymicrobial infection where beneficial vaginal bacteria, mainly hydrogen peroxide and lactic acid-producing *Lactobacillus* species, which are usually dominant in vaginal microbiota of healthy women, are replaced by high concentrations of facultative and strict anaerobic bacteria [1,9]. The most prominent of these are *Gardnerella* spp., facultative anaerobes usually found embedded in a polymicrobial BV biofilm [10,11,12]. However, *Gardnerella* spp. are also commonly found in asymptomatic or BV-negative women [13]. This has aroused interest in whether genetic differences among *G. vaginalis* isolates might differentiate pathogenic from commensal organisms [14]. *G. vaginalis* was the only recognized species in its genus for over four decades, but very recently the *Gardnerella* taxonomic description was amended based on comparisons of whole-genome sequencing and matrix-assisted laser desorption/ionization time-of-flight (MALDI-TOF) mass spectrometry analysis, resulting in four species (*G. vaginalis*, *G. leopoldii*, *G. swidsinskii*, and *G. piotii*) and nine ‘genome species’ [15]. It is likely that each named *Gardnerella* species and ‘genome species’ are specifically associated with either BV or healthy vaginal microbiota due to differences in their virulence potential, as suggested by several studies [16,17,18,19,20,21]. Following this renewed taxonomy of the genus *Gardnerella*, in this article, the term *Gardnerella* spp. will be used to address previous publications, since we cannot rule out the fact that other *Gardnerella* species were involved.

While BV etiology is still a matter of debate [1], a common hypothesis suggests that virulent strains of *Gardnerella* spp. initiate the formation of the biofilm on vaginal epithelial cells and become a scaffolding to which other BV-associated anaerobes thereafter can attach [9]. *Atopobium vaginae* is one of the species that is often found associated with *Gardnerella* spp. biofilms [22]. Evidence suggests that the therapeutic failures and recurrences of BV might be associated with the presence of high loads of *A. vaginae*, since this species presents specific resistance against standard antibiotics [23,24,25]. Additionally, *A. vaginae* has been positively associated with vaginal discharge in women with BV, an elevated vaginal pH, and the presence of clue cells [26]. It was also described that high vaginal loads of *A. vaginae* in combination with *Gardnerella* spp. are related to late miscarriage and pre-maturity [23].

Recently it has been pointed out that *Prevotella bivia* might have an important role in the early stage of BV development [9,27]. Using daily vaginal swabs, Muzny and colleagues found that *P. bivia* was the first BV-associated species to increase in relative abundance above baseline prior to incident BV, followed shortly thereafter by an increase in the relative abundance of *Gardnerella* spp., suggesting that synergism between *P. bivia* and *Gardnerella* spp. might be an important event prior to BV [28]. In fact, an earlier in vitro study demonstrated that *Gardnerella* spp. and *P. bivia* can act synergistically [29]. The authors showed that *Gardnerella* spp. produce amino acids through their metabolism, which can be used by *P. bivia* as its nutrient source which results in the production of ammonia, which in turn is used by *Gardnerella* spp. More recently, Gilbert and colleagues established an in vivo BV model in which they coinfected mice with *Gardnerella* spp. and *P. bivia*, revealing that *Gardnerella* facilitates ascension of *P. bivia* into the uterine horns [30]. The virulence potential of *P. bivia* is also derived from studies that associated its colonization with preterm birth, endometritis, and other uterine pathologies [31,32,33]. 

Knowledge of the microbial interactions in the vaginal ecosystem during BV is still scarce since functional microbial studies in polymicrobial biofilms are very limited [34]. We have recently shown that when growing dual-species BV biofilms, distinct microbial interactions can occur, including antagonistic [35] or synergistic biofilm accumulation [35,36], as well as molecular interactions that have an impact on *Gardnerella* gene expression [37]. Thus, in this study, we aimed to develop and characterize, for the first time, an in vitro model containing *G. vaginalis*, *A. vaginae*, and *P. bivia*, and to investigate the ability of these three species to form a multi-species biofilm, with particular attention to their ability to induce alterations in key genes of interest. 

## 2. Results

### 2.1. Quantification of the Biomass of Mono-, Dual-, and Triple-Species BV-Associated Biofilms 

In the BV-associated vaginal ecosystem, resident microorganisms interact with each other in both synergistic and antagonistic manners, which might affect their ability to form biofilms in this polymicrobial community [35]. Initially, we compared the in vitro biofilm formation ability of each tested species for 24 h and 48 h using optimized [38] in vitro conditions (Figure 1A). In both timepoints tested, *G. vaginalis* formed a biofilm with the highest total biomass, while *P. bivia* formed a biofilm with the lowest total biomass. Following the hypothesis that *Gardnerella* is the early colonizer in BV [9], we then assessed how *A. vaginae* and *P. bivia* could incorporate into a 24-h pre-formed *G. vaginalis* biofilm. Under the tested conditions, all the consortia reached the same level of total biofilm biomass (Figure 1B). Curiously, when compared to the mono-species *G. vaginalis* 48-h biofilm, none of the consortia provided an added advantage in terms of an increase in total biomass. While it was tempting to compare the 48-h consortia with *A. vaginae* or *P. bivia* 48-h mono-species biofilms, it is important to highlight that both species were only allowed to grow for 24 h, after the initial 24-h *G. vaginalis* biofilm was performed, as described in the methods section. 

### 2.2. In Vitro PNA Gard162 and PNA AtoITM1 Probes Specificity and Efficiency

Although the PNA Gard162 [39] and PNA AtoITM1 [40] probes’ specificity had been previously tested for several BV-associated bacteria, we also analyzed these probes’ specificity for the bacterial strains used in this study. Based on our results (Table 1), PNA Gard162 and PNA AtoITM1 probes hybridized with *G. vaginalis* and *A. vaginae*, respectively, whereas no hybridization was observed for the other species tested, showing a specificity of 100% as previously reported. Additional details of the specificity of the Gard162 and AtoITM1 probes are shown in Appendix A. 

As no *P. bivia* PNA FISH probe currently exists, the estimation of *P. bivia* counts could only be assessed indirectly by 4′-6-diamidino-2-phenylindole (DAPI) counterstaining, assuming that all cells with unlabeled PNA probes were *P. bivia;* however, this needs to be experimentally determined [41]. As such, we compared the data obtained from PNA FISH and DAPI counts for both *G. vaginalis* and *A. vaginae* pure-culture biofilms and the planktonic fractions of the biofilm. Not surprisingly, each probe failed to detect 100% of the respective total cells. By performing serial dilutions of each sample, calibration curves were obtained for *G. vaginalis* biofilm (Figure 2A) or planktonic cells (Figure 2B) and for *A. vaginae* biofilm (Figure 2C) or planktonic cells (Figure 2D).

Taking into consideration this data, it was possible to calculate the efficiency of each probe and to develop an equation that would correct PNA counts, to prevent overestimation of DAPI counts as *P. bivia* counts, as listed in Table 2.

### 2.3. Quantification and Distribution of Bacterial Populations in Dual- and Triple-Species BV-Associated Biofilms by PNA FISH

Taking advantage of the robustness of the PNA FISH/DAPI approach for the differentiation between *G. vaginalis*, *A. vaginae,* and *P. bivia,* we discriminated the bacterial populations into dual- and triple-species BV-associated biofilms and their planktonic fractions. Similar to what was described before [42], we showed that *A. vaginae* and *P. bivia* were able to incorporate in the dual-species biofilms, accounting for up to ∼23% and ∼38% of the total number of cells, respectively (Figure 3). Curiously, in the triple-species biofilms, the relative load of *A. vaginae* was reduced to ~8.3%, maintaining *G. vaginalis* as the main species. The lower ability of *A. vaginae* to integrate the triple-species biofilms is evidence that somewhat specific interactions are established when these three bacterial species act as a consortium, which in our tested conditions promoted the enhanced integration of *P. bivia*, in depreciation of *A. vaginae*. When looking at the bacterial populations found on the planktonic fraction of the biofilm, it is noteworthy that significant differences were found both for *P. bivia* and the triple-species consortia; in contrast the relative composition of both biofilms and planktonic fractions were similar for *A. vaginae*.

We also analyzed bacterial distribution in the intact structure of the dual and triple-species biofilms by confocal laser scanning microscopy (CLSM). As shown in Figure 4, in both dual- and triple-species biofilms, *A. vaginae* and *P. bivia* were found well-distributed across the *G. vaginalis* biofilm, in small clusters of cells. Details in the orthogonal views of mono-, dual-, and triple- species biofilms are shown in Appendix A.

### 2.4. Impact of A. vaginae and P. bivia on G. vaginalis Virulence

Considering the central role often attributed to *Gardnerella* in BV etiology [1,43], shifts in the *G. vaginalis* transcriptome during the establishment of polymicrobial BV-associated biofilms could be a key for unveiling interspecies interactions that enhance the virulence of *G. vaginalis.* Thus, we focused on deciphering the impact of *A. vaginae* and/or *P. bivia* on *G. vaginalis* virulence. As such, we analyzed the expression of genes related to cytotoxicity, biofilm maintenance, antimicrobial resistance, and evasion of the immune system in cells from mono-, dual-, and triple-species biofilms. Regarding cytotoxicity, *G. vaginalis* produces the toxin vaginolysin (*vly*), which might induce lysis in vaginal cells membranes [44,45]. However, as shown in Figure 5A, according to our in vitro conditions, no significant differences were found in the expression of this gene by *G. vaginalis* between the different biofilm models. We also addressed sialidase (*sld*) expression, since sialidase appears to contribute to *G. vaginalis* cytotoxicity by the destruction of the protective mucus layer on the vaginal epithelium [46]. Although we detected a slight decrease in *sld* transcription, it was not statistically significant in dual- and triple-species biofilms, as compared with mono-species *G. vaginalis* biofilms (Figure 5B).

Regarding the glycosyltransferases, it has been proposed that they are involved in the transfer of a sugar moiety to a substrate and are thus essential in the biosynthesis of glycoconjugates like exopolysaccharides and glycoproteins, which are important for biofilm maintenance to maximize the full virulence of *G. vaginalis* [47,48]. Of note, the expression of the *HMPREF0424_0821* transcript, which encodes glycosyltransferases type II, was up-regulated in all the tested conditions, being approximately 2.8-fold higher in triple-species than in mono-species biofilms (*p* < 0.05; Figure 5C). We also analyzed the expression of transcripts encoding antimicrobial-specific resistance proteins belonging to efflux pump families (*HMPREF0424_1122* and *HMPREF0424_0156).* Despite detecting slight changes in the transcription of these genes, these were not statistically significant (Figure 5D,E). Nevertheless, a tendency was observed in the presence of *P. bivia*, with a downregulation of the tested *G. vaginalis* genes, similar to what we have previously found [37]. Finally, we analyzed the expression of *HMPREF0424_1196* transcript, which encodes a Rib-protein that belongs to the α-like protein (Alf)-family of highly repetitive surface antigens [49], which elicit protective immunity through their inter-strain size variability [48]. Here the most striking difference was in the triple-species biofilm, as compared to the *G. vaginalis* mono-species biofilm (*p* < 0.05; Figure 5F). 

## 3. Discussion

The dynamic and complex nature of the vaginal microbiota and the likely role of multiple bacterial species in the pathogenesis of BV have posed major challenges for developing realistic polymicrobial in vitro biofilm models [50,51]. To date, the majority of in vitro studies only address mono- or dual-species biofilms, and are focused on *Gardnerella* species [37,52]. Herein, we describe, for the first time, an in vitro biofilm composed of *G. vaginalis*, *A. vaginae,* and *P. bivia*, three highly relevant BV-associated species [28,30]. In a polymicrobial community, bacterial species interact extensively with each other and these interactions might also determine the structure and composition of multi-species biofilms [53]. It is, thus, reasonable to assume that in triple-species BV-associated biofilms the microbial interactions become more complex than in dual- or mono-species biofilms [54]. 

Our experimental model follows the hypothesis that *Gardnerella* is the early colonizer in BV [9], to which we later allowed co-incubation with *A*. *vaginae* and *P. bivia.* In vitro experimental data supporting the role of some *Gardnerella* species to be the early colonizers is derived from the fact that some *Gardnerella* isolates have a significantly higher ability to adhere to epithelial cells than many other BV-associated species [52,55,56] and are also able to displace vaginal lactobacilli [57], a pivotal step in the development of a characteristic multi-species biofilm. It is also important to bear in mind that the recent reclassification of *G. vaginalis* [15] in multiple *Gardnerella* species might have important implications in BV etiology, as a particular species might have a different role in BV development, although this still needs to be further explored [19,21,58]. Herein, we selected a type strain of *G. vaginalis* since its complete genome is available, and thus, it is well genotypically characterized, being used in several studies associated with BV [15,58]. Regarding other BV-associated species, we selected *A. vaginae* and *P. bivia* to be included in this study, due to data suggesting they may be more than bystanders [9,32]. Indeed, in a prospective vaginal microbiome study Muzny and colleagues showed that the mean relative abundance of *P. bivia*, *Gardnerella* spp., and *A. vaginae* became significantly higher in cases four days before (*P. bivia*), three days before (*Gardnerella* spp.), and on the day of (*A. vaginae*) incident BV onset [28]. Based on this study, the authors suggested that together with virulent *G. vaginalis*, *P. bivia*, and *A. vaginae* may have potential key roles in the induction and development of incident BV [9]. Nevertheless, knowledge about the microbial relationships between these three bacterial species remains scarce. In order to shed new light on this aspect, we analyzed the effect of *A. vaginae* and *P. bivia* on biofilm formation and its impact on *G. vaginalis* pathogenicity. 

Similar to what is described in oral infections, in which the interactions between microbial communities have tremendous importance for the development of oral disease [59], we hypothesize that *G. vaginalis*, *A. vaginae,* and *P. bivia* establish a network of interactions that affect the development of the BV-associated biofilm, a key hallmark of BV. Curiously, under the tested conditions, inoculation or co-inoculation of the different species on the pre-formed *G. vaginalis* biofilm did not significantly enhance the amount of total biofilm biomass. However, despite widespread utilization of the CV staining method for biofilm quantification used in this study, an inherent limitation of this method is that total biomass comparison between species is not possible, since different species produce distinct biofilm matrices and have different cell sizes [60,61]. Thus, interpretation of these data should be made with reservation. Nevertheless, as shown in our CLSM and PNA-FISH quantitative data, dual- and triple-species biofilms contained significant numbers of each species, with *G. vaginalis* as the most prominent species, similar to in vivo studies [11,22]. Interestingly, our PNA FISH data suggest that different microbial relationships are established in dual- and triple-species biofilms. This is inferred by the distinct bacterial composition observed in multi-species biofilms, in particular the triple-species biofilm where the relative composition of *G. vaginalis* and *P. bivia* increased while *A. vaginae* decreased. This is even more relevant since *P. bivia* formed a weak mono-species biofilm and grew slower than *A. vaginae* and *G. vaginalis* in New York City III (NYC III) medium, as shown previously [38]. It is important to mention that several factors influenced the bacterial integration in biofilm, including the growth rates of the bacterial species [62] and the ability to adhere to a surface [63,64] and to each other [65,66]. Interestingly, despite the slower growth rate, *P. bivia* was better fit to grow in the biofilm than *A. vaginae* in our tested conditions. This suggests that *G. vaginalis* modifies the local environment, making it more favorable for the growth of *P. bivia*, which might be a result of an ammonia flow mechanism as proposed by Pybus and colleagues [29].This further demonstrates that different bacterial species influence the growth of other species, and likely have an impact in BV etiology. It is also interesting to observe that the planktonic fraction of the dual- and triple-species biofilm did not always coincide with the biofilm consortia, suggesting that the specific microenvironment of the biofilms provide different competitive advantages, as shown elsewhere [67,68]. 

It is noteworthy that synergistic effects were, however, observed when we analyzed changes in *G. vaginalis* gene expression. The expression of the *HMPREF0424_0821* transcript, which encodes a glycosyltransferase, was significantly up-regulated in the multi-species consortia. Glycosyltransferases have been described as key enzymes required for biofilm maintenance [48], and may also have a putative role for cell surface glycoconjugates which has been proposed to shape vaginal microbiota–host interactions [69]. This enhancement was only significant in the triple-species biofilm model, highlighting, therefore, the importance of the interplay between multiple bacterial species in the development of BV. Studies to assess the role of glycosyltransferase on biofilm formation and virulence have been conducted for other species, namely for *Streptococcus mutans*, a bacterium responsible for the initiation of tooth decay [70]. It was shown that glycosyltransferase mediated biofilm matrix dynamics and virulence. Interestingly, the deletion of the glycosyltransferase gene resulted in no change in overall biofilm biomass, however, the mutant strain originated an altered biofilm architecture. Concurrently, the mutant was less virulent in an in vivo rat model of dental caries [70]. Corroborating the relevance of glycosyltransferase in biofilms, we have recently shown that this gene was up-regulated in *Gardnerella* spp. in 15 different dual-species consortia [37]. Together, these data emphasize the importance of other BV-associated species in *G. vaginalis* virulence, and consequently, in host infection.

While we have previously shown that *P. bivia* could enhance *G. vaginalis vly* expression two-fold, when a dual-species biofilm was grown in supplemented brain-heart infusion (sBHI) medium [37], in NYC III medium *vly* expression was not changed. Regarding the role of *sld*, some early studies pointed out that this gene is strongly linked with the development of a biofilm [71,72], however, in our tested conditions, no relevant changes were verified when *G. vaginalis* was cultivated in mono- or multi-species biofilms, corroborating our previous findings [37]. Such evidence supports a recent study that postulated that *sld* does not likely have a role in establishing or maintaining the biofilm [73]. It should be pointed out that in this study we chose to use NYC III medium, since sBHI was not appropriate to induce mono-species biofilms from either *P. bivia* or *A. vaginae* [38]. Since bacterial gene expression is strongly influenced by media conditions [74,75], we cannot exclude that the optimal growth conditions provided by NYC III could also somehow be affecting gene expression by *G. vaginalis.* Indeed, by comparing the data from this study with our previous findings [37], we observed that using NYC III, the base level of *G. vaginalis vly* expression was approximately 10-fold higher than in comparison with sBHI, which could explain the differences observed in both studies. On account of the fact that synergistic effects often occur when one bacterium is providing some advantage to another [76,77], by using an optimal biofilm-inducing media such as NYC III, it might be possible that some synergistic effects were masked. Evidence to support this hypothesis can be observed by the fact that several strains of *A. vaginae* easily die out when grown in sBHI medium, but maintain viability when co-cultured with *G. vaginalis* [78]. However, in NYC III, *A. vaginae* is able to survive alone. Furthermore, for the other tested genes, with exception of *sld*, we also noted a higher base level (2- to 5-fold) of expression in NYC III as compared with sBHI. Despite these differences, a similar effect on *G. vaginalis* transcriptomic profile was observed for the remaining tested genes, as compared with our previous dual-species biofilm study [37]. However, further studies are required to elucidate the influence of the growth media in gene expression by *G. vaginalis*. In triple-species biofilms, we observed a downregulation of these genes when compared to *G. vaginalis* mono-species biofilms, however, this requires future study to understand the molecular mechanisms involved in antimicrobial resistance and evasion of the immune system. Taken together, our data suggest that microbial relationships between co-infecting bacteria can influence the development of a polymicrobial biofilm, a marker of BV. However, more research is needed to provide a better mechanistic insight into the complex interplay between *G. vaginalis*, *A. vaginae*, and *P. bivia*, and their eukaryotic hosts. While the choice of a very rich medium was used to guarantee that all the tested bacteria were able to grow in vitro in a biofilm phenotype, trying to determine these microbial interactions in conditions more similar to the vaginal environment might provide novel insights, as we previously demonstrated that growing *Gardnerella* spp. in a medium simulating genital tract secretions and complemented with components of the host immune system had a significant impact on the growth and biofilm formation [79]. Understanding the molecular basis and biological effect of these microbial interactions and microbial–host interactions may provide novel information necessary to define more effective and goal-oriented treatment in BV and improve women’s reproductive health.

## 4. Materials and Methods 

### 4.1. Bacterial Strains and Culture Conditions

*G. vaginalis* strain ATCC 14018^T^, *A. vaginae* strain ATCC BAA-55^T^, and *P. bivia* strain ATCC 29303^T^ were used in this study. Each inoculum was grown in New York City III broth (NYC III) ((1.5% (*w*/*v*) Bacto™ proteose peptone no. 3 (BD, Franklin Lakes, NJ, USA), 0.5% (*w*/*v*) glucose (Thermo Fisher Scientific, Lenexa, KS, USA), 0.24% (*w*/*v*) HEPES (VWR, Sparks, NV, USA), 0.5% (*w*/*v*) NaCl (VWR), and 0.38% (*w*/*v*) yeast extract (Liofilchem, Roseto degli Abruzz, Italy)) supplemented with 10% (*v*/*v*) inactivated horse serum (Biowest, Nuaillé, France) [38] for 24 h at 37 °C under anaerobic conditions (AnaeroGen Atmosphere Generation system, Oxoid, Hampshire, United Kingdom), as we previously showed this to be the optimal condition to grow mono-species biofilms of the selected bacterial species [38].

### 4.2. Biofilm Formation and Biomass Quantification by the Crystal Violet Method

Dual- and triple-species biofilms were initiated by inoculating a 10^7^ colony-forming units (CFU)/mL bacterial suspension of *G. vaginalis* into 24-well tissue culture plates (Orange Scientific, Braine L’Alleud, Belgium) and by incubating the plates for 24 h, at 37 °C, and under anaerobic conditions. After 24 h, planktonic cells were removed, and 10^7^ CFU/mL of each bacterial species, *A. vaginae* or *P. bivia* (for dual-species biofilms) and *A. vaginae*, *P. bivia* (for triple-species biofilms), were inoculated in the pre-formed *G. vaginalis* biofilms and incubated for another 24 h. Of note, we first adjusted the bacterial concentration of the bacterial suspension to 9 × 10^7^ CFU/mL due to the limit of detection of the microplate reader and then we diluted to 1 × 10^7^ CFU/mL, confirming this concentration by CFUs. At 620 nm, 9 × 10^7^ CFU/mL of *G. vaginalis* corresponds to an optical density (OD) = 0.15; for *A. vaginae* an OD = 0.11, and for *P. bivia* an OD = 0.16. Mono-species biofilms of *G. vaginalis* were grown as a control for 48 h, in which fresh medium was added to the respective wells after the first 24 h of biofilm formation. In addition, we also performed 24-h mono-species biofilm growth for *A. vaginae* and *P. bivia* to examine their individual ability to grow in the tested conditions. To quantify the biomass of mono-, dual-, and triple-species biofilms, we used the crystal violet (CV) method [80]. In brief, after the fixation step with 100% (*v*/*v*) methanol (Thermo Fisher Scientific) for 20 min, biofilms were stained with CV solution at 1% (*v*/*v*) (Merck, Darmstadt, Germany) for 20 min. Each well was washed twice with phosphate-buffered saline, and bound CV was released with 33% (*v*/*v*) acetic acid (Thermo Fisher Scientific). To estimate total biofilm biomass, the OD of the resulting solution was measured at 595 nm. Biofilm assays were repeated three times on separate days, with four technical replicates assessed each time.

### 4.3. In Vitro PNA Gard162 and AtoITM1 Probes Specificity and Efficiency

Although the specificity of PNA Gard162 [39], and AtoITM1 [40] probes have been previously tested, we also evaluated the probe specificity for the bacterial species used in this study. Thus, we performed an experiment in order to detect any possible cross-hybridization. The evaluation of PNA FISH hybridization was based on a qualitative score, as previously described [39]: (−) absence of hybridization, (++) moderate hybridization, (+++) good hybridization, and (++++) optimal hybridization. 

We then carried out an experiment to analyze the efficiency of both PNA probes. As such, we performed several dilutions from pure bacterial suspensions obtained from mono-species biofilms and their planktonic fractions. To determine the efficiency of each probe, the same sample was hybridized with a species-specific probe and then stained with 4′-6-diamidino-2-phenylindole (DAPI, 2.5 μg/mL) to account for nonhybridizing bacteria; defined as double staining. After the double staining, the bacteria were enumerated at two different wavelengths at the same position within the sample. Based on both data, we performed a correlation between the PNA counts and the DAPI counts that allowed us to obtain the equations shown in Figure 2. All experiments were performed in triplicate.

### 4.4. Quantification of Bacterial Populations in Dual- and Triple-Species Biofilms and Their Planktonic Fraction by PNA FISH

The bacterial population within the 48-h multi-species biofilms and in their planktonic fraction was discriminated using the peptide nucleic acid fluorescence in situ hybridization (PNA FISH) method, as previously described [37]. Briefly, after fixing the biofilm suspension, a PNA probes specific for *G. vaginalis* (Gard162) and for *A. vaginae* (AtoITM1) were added to each well of epoxy-coated microscope glass slides (Thermo Fisher Scientific). An additional staining step was done at the end of the hybridization procedure, covering each glass slide with DAPI (2.5 μg/mL). Microscopic visualization was performed using filters capable of detecting the PNA Gard162 probe (BP 530-550, FT 570, LP 591 sensitive to the Alexa Fluor 594 molecule attached to the Gard162 probe), the PNA AtoITM1 probe (BP 470–490, FT500, LP 516 sensitive to the Alexa Fluor 488 molecule attached to the AtoITM1 probe), and DAPI (BP 365–370, FT 400, LP 42). Twenty fields were randomly acquired in each sample. The number of bacteria was counted using *ImageJ Software* [59]. To reduce any possible overestimation due to the use of DAPI as the probe efficiency was not 100%, we then applied the equations from Table 2 to obtain more accurate relative values. These assays were repeated three times on separate days.

### 4.5. Confocal Laser Scanning Microscopy Analysis of Biofilm Bacterial Distribution 

To analyze the bacterial distribution of dual- and triple-species biofilms, the biofilm structure was evaluated by confocal laser scanning microscopy (CLSM) using the PNA Gard162 and AtoITM1 probes coupled to DAPI staining, as described above. For this experiment, biofilms were formed on an 8-well chamber slide (Thermo Fisher Scientific™ Nunc™ Lab-Tek™, Bohemia, NY, USA) at 37 °C under anaerobic conditions for 48 h with the replacement of NYC III medium at 24 h of growth and the addition of the respective BV-associated bacteria. The CLSM images were acquired in an Olympus™ FluoView FV1000 (Olympus, Tokyo, Japan) confocal scanning laser microscope, using a 40× objective. Microscopic visualization was performed using lasers capable of detecting the PNA Gard162 probe (Laser 559, excitation wavelength 559 nm, emission wavelength 618 nm, BA 575–675, sensitive to the Alexa Fluor 594 molecule attached to the Gard162 probe), the PNA AtoITM1 probe (Laser 488, excitation wavelength 488 nm, emission wavelength 520 nm, BA 505–540, sensitive to the Alexa Fluor 488 molecule attached to the AtoITM1 probe), and DAPI (Laser 405, excitation wavelength 405 nm, emission wavelength 461 nm, BA 430–470). Images were acquired with 640 × 640 resolutions of each surface analyzed. All assays were repeated three independent times with two technical replicates.

### 4.6. G. vaginalis Gene Expression Quantification in Mono-, Dual-, and Triple-Species Biofilms

Gene expression of six potential *Gardnerella* virulence genes, specifically vaginolysin (*vly*), sialidase (*sld*), glycosyltransferase, type II (*HMPREF0424_0821*), multidrug ABC transporter (*HMPREF0424_1122*), bacitracin transport, ATP-binding protein BcrA (*HMPREF0424_0156*), and a transcript that encodes a Rib-protein (*HMPREF0424_1196*) was determined in 48-h mono-, dual-, and triple-species biofilms. Total RNA was extracted using an E.Z.N.A.^®^ Bacterial RNA Kit (Omega Bio-tek, Norcross, GA, USA) with minor changes, as previously optimized [81]. Next, genomic DNA was degraded with one step of DNase treatment (Fermentas, Vilnius, Lithuania) following the manufacturer’s instructions. RNA concentration, purity, and integrity were determined, as previously described [82]. The same amount of total RNA (300 ng/μL) was reverse transcribed using the RevertAid™ First Strand cDNA synthesis kit (Fermentas), as previously optimized [83], and gene-specific reverse transcription primers as a priming strategy. Quantitative PCR (qPCR) was prepared by mixing 5 µL of iQ SYBR green supermix (Bio-Rad, Hercules, CA, USA), 2 µL of 1:100 diluted cDNA, 0.5 µL of 5 µM forward and reverse primers (Appendix A), and water up to 10 µL. The run was performed in a CFX96^TM^ thermal cycler (Bio-Rad) with the following cycling parameters: 3 min at 95 °C, followed by 45 cycles of 10 s at 95 °C, 10 s at 60 °C, and 15 s at 72 °C. Reaction efficiency was determined by the dilution method [84]. It is of note that at 60 °C, all sets of primers used (Appendix A) had similar efficiencies. In addition, the analysis of the melting curves confirmed the presence of a single peak, providing evidence for the specificity of the tested primers. Normalized gene expression was determined by using the delta C_t_ method (E^ΔCt^), a variation of the Livak method, where ΔC_t_ = C_t_ (reference gene) − C_t_ (target gene) and E stands for the reaction efficiency experimentally determined. A non-reverse transcriptase control was included in each reaction. All assays were repeated at least three independent times with three technical replicates.

### 4.7. Statistical Analysis

The data were analyzed using the statistical package GraphPad Prism version 6 (La Jolla, CA, USA) by paired *t*-test, two-way ANOVA (Sidak’s multiple comparison test) and Mann–Whitney U test for the data that did not follow a normal distribution according to the Kolmogorov–Smirnov test. Values with *p* < 0.05 were considered statistically significant. 

## Figures and Tables

**Figure 1 pathogens-10-00247-f001:**
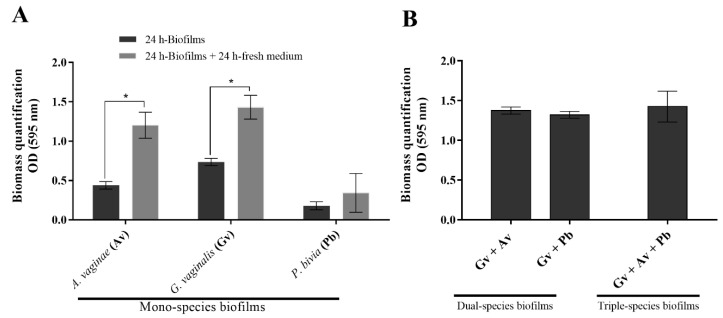
Total biomass of mono- and multi-species bacterial vaginosis (BV)-associated biofilms was determined by staining with crystal violet (CV). (**A**) Total biofilm biomass of 24-h and 48-h mono-species biofilms for the three microorganisms of interest. (**B**) Total biofilm biomass of dual- and triple-species BV-associated biofilms at 48 h. Dual- and triple-species biofilms were initiated by inoculating a bacterial suspension of *G. vaginalis* into 24-well tissue culture plates in New York City III (NYC III) medium and by incubating the plates for 24 h, at 37 °C under anaerobic conditions. After 24 h, planktonic cells were removed, and each bacterial species, *Atopobium vaginae* or *Prevotella bivia* (for dual-species biofilms) and *A. vaginae* and *P. bivia* (for triple-species biofilms), were inoculated in the pre-formed *G. vaginalis* biofilms and incubated for another 24 h. Each data point represents the mean ± s.d. of three independent assays, with four technical replicates assessed each time. * Values were significantly different between 24-h and 48-h mono-species biofilms (paired *t*-test, *p* < 0.05). Abbreviations: *A. vaginae* (Av), *G. vaginalis* (Gv), and *P. bivia* (Pb).

**Figure 2 pathogens-10-00247-f002:**
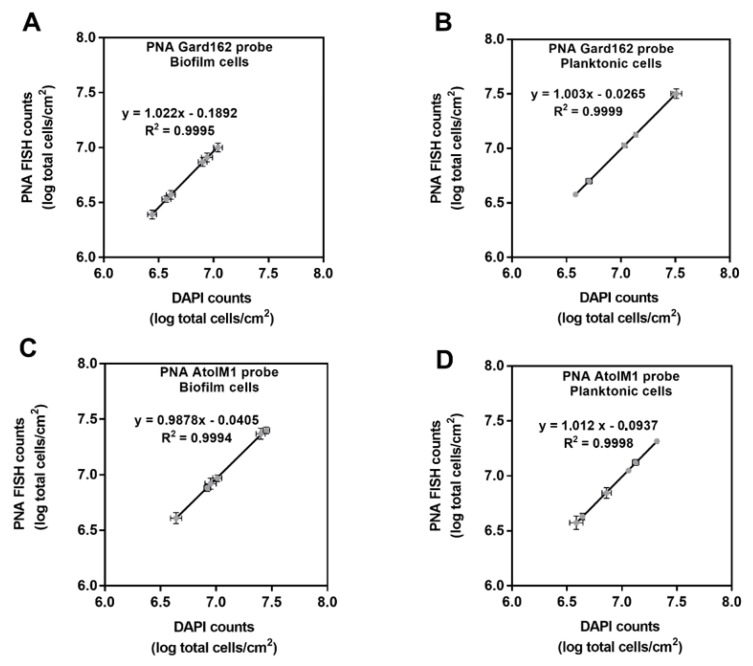
Correlation between PNA FISH counts and DAPI counts for mono-species biofilms and for their planktonic fraction at different bacterial concentrations. (**A**) *G. vaginalis* biofilm cells that were identified indirectly by DAPI coincided with the populations quantified by PNA FISH using PNA Gard162 probe. (**B**) *G. vaginalis* planktonic cells that were identified indirectly by DAPI coincided with the populations quantified by PNA FISH using PNA Gard162 probe. (**C**) *A. vaginae* biofilm cells that were identified indirectly by DAPI coincided with the populations quantified by PNA FISH using PNA AtoITM1 probe. (**D**) *A. vaginae* planktonic cells that were identified indirectly by DAPI coincided with the populations quantified by PNA FISH using PNA AtoITM1 probe. Each data point represents the mean ± s.d. from three independent assays. For each assay, 20 fields were randomly acquired in each sample and the number of bacteria per image was counted using *ImageJ Software*.

**Figure 3 pathogens-10-00247-f003:**
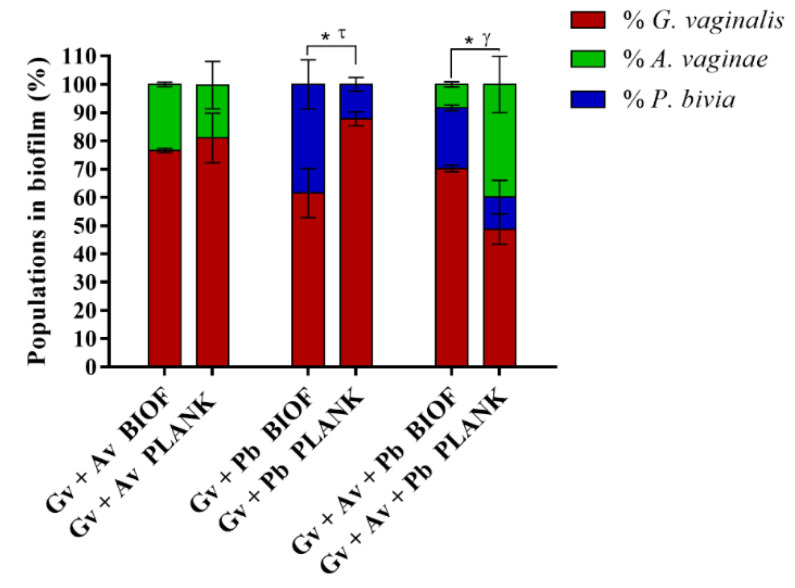
Bacterial populations in dual- and triple-species BV-associated biofilms and in their respective planktonic fraction. Biofilms were disrupted and resuspended before quantification was performed, as described in methods section. Percentage of cells detected by PNA FISH for 48-h biofilms and in their planktonic fraction. Each data point represents the mean ± s.d. of three independent assays. For each assay, 20 fields were randomly acquired in each sample and the number of bacteria per image was counted using *ImageJ Software.* Values were significantly different between the percentage of each bacterial species that integrates the dual- or triple- species biofilm or planktonic fraction, namely, * Gv BIOF vs. Gv PLANK, ^γ^ Av BIOF vs. Av PLANK, ^τ^ Pb BIOF vs. Pb PLANK (two-way ANOVA test, *p* < 0.05). Abbreviations: *A. vaginae* (Av), *G. vaginalis* (Gv), and *P. bivia* (Pb); biofilm BIOF; planktonic (PLANK).

**Figure 4 pathogens-10-00247-f004:**
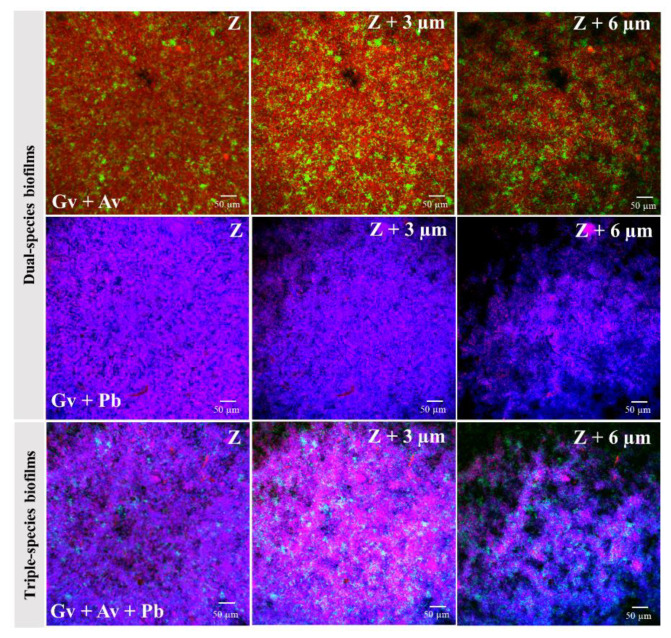
An example data set on the organization of the dual- and triple-species BV-associated biofilms by confocal laser scanning microscopy (CLSM). Gv and Av cells were differentiated by hybridization with PNA Gard162 (red/purple color when coupled with DAPI) and AtoITM1 probes (green/blue-green color when coupled with DAPI), respectively, while Pb was differentiated by DAPI (blue color). Abbreviations: *A. vaginae* (Av), *G. vaginalis* (Gv), and *P. bivia* (Pb).

**Figure 5 pathogens-10-00247-f005:**
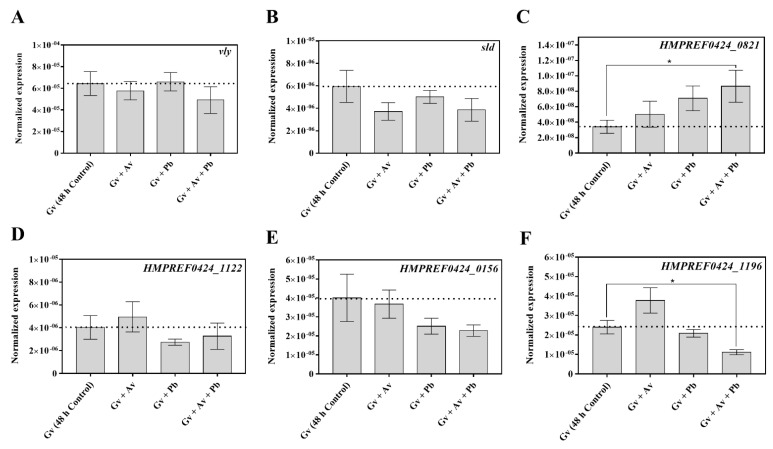
Quantification of the transcription of known virulence genes in *G. vaginalis* cultured under mono-, dual-, and triple-species biofilms. (**A**) Quantification of vaginolysin (*vly*) transcription. (**B**) Quantification of sialidase (*sld*) transcription. (**C**) Quantification of *HMPREF0424_0821* transcript, which encodes type II glycosyl-transferase. (**D**) Quantification of *HMPREF0424_1122* transcript, which encodes a multidrug ABC transporter. (**E**) Quantification of *HMPREF0424_0156* transcript, which encodes bacitracin transport, ATP-binding protein BcrA. (**F**) Quantification of *HMPREF0424_1196* transcript, which encodes a Rib-protein. For qPCR experiments, the bars represent the mean, and the error bars the standard error of the mean (mean ± s.e.m.) of at least three independent assays. * Values are significantly different between the triple-species consortium and the *G. vaginalis* mono-species biofilm under the same conditions (non-parametric Mann–Whitney U test, *p* < 0.05). Abbreviations: *A. vaginae* (Av), *G. vaginalis* (Gv), and *P. bivia* (Pb).

**Table 1 pathogens-10-00247-t001:** Bacterial species used in peptide nucleic acid fluorescence in situ hybridization (PNA FISH) assays and their specificity with PNA probes ^a^.

Strains	Gard162 Probe Specificity	AtoITM1 Probe Specificity
*G. vaginalis* strain ATCC 14018^T^	++++	−
*A. vaginae* strain ATCC BAA-55^T^	−	++++
*P. bivia* strain ATCC 29303^T^	−	−

^a^ PNA probes’ (Gard162 and AtoITM1) specificity was tested for each species, with the following hybridization PNA FISH qualitative evaluation: (−) absence of hybridization; (++++); and optimal hybridization.

**Table 2 pathogens-10-00247-t002:** Equations used to quantify the bacterial population in biofilms cells and their planktonic fraction.

Condition	Equation	PNA Probe Efficiency (%)
*G. vaginalis* biofilm	*G. vaginalis* counts = (log (PNA Gard162 probe bacterial counts/area) + 0.1892)/1.022	92.08
*A. vaginae* biofilm	*A. vaginae* biofilm cells counts = (log (PNA AtoITM1 probe bacterial counts/area) + 0.0405)/0.9878	91.59
Planktonic fraction of *G. vaginalis* biofilm	*G. vaginalis* planktonic cells counts = (log (PNA Gard162 probe bacterial counts/area) + 0.0265)/1.003	98.67
Planktonic fraction of *A. vaginae* biofilm	*A. vaginae* planktonic cells counts = (log (PNA AtoITM1 probe bacterial counts/area) + 0.0937)/1.012	98.12

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
