# Peer review of "Atopobium vaginae and Prevotella bivia Are Able to Incorporate and Influence Gene Expression in a Pre-Formed Gardnerella vaginalis Biofilm"

_pathogens, 2021, doi:10.3390/pathogens10020247_

Round 1
Reviewer 1 Report
This manuscript is well-written with thorough background information. As preliminary data examining dual and triple species biofilms, it begins to address an important question about the role of multi-species interactions in vaginal biofilms leadings to BV. There are a few essential experiments that should be included to ensure that the data are significant to the field.
Major concerns:
One essential experiment to include would be to compare the numbers from Figure 3 with CFU counts on these biofilms. If only about 90% of the cells are being properly stained, that could mean a high overestimation of P. bivia populations in the biofilm. Without CFU numbers, it is impossible to tell. While the PNA Fish is helpful for visualization, it is not the most accurate for determining biomass. These counts are essential. It appears as though some of the G. vaginalis may be getting displaced by the other organisms in the final 48 our biofilm as the counts of G. vaginalis must be higher in the single species biofilms than the multi-species. This again could be better teased out by including CFU counts in the paper.
Why do the authors not present more information on the biofilm architecture? If these are confocal images, side stacks would be useful to see what the authors are talking about when they say that there are small clusters of different species. Without those images or spatial analysis, the readers aren’t able to verify this.
A larger scale transcriptomics approach would have been highly preferrable to a few genes examined via qPCR but of course that is not always possible with available funds. That would have made the paper stronger though and should be considered in the future (as I’m sure it was by the authors).
Minor:
- It would help the reader to briefly mention in the early results section of the paper how these biofilms were grown (surface and medium) as well as the addition of the two species to pre-formed biofilms. It is discussed in the methods but a brief summary in the results or even in the figure legends would be helpful.
- One additional sentence on how the probe efficiency was used to derive the equations used in lines 122-123 would be helpful to the reader.
- Can the authors comment on why they think the probe efficiency was lower in biofilms than in their planktonic cells? Is it a penetration issue?
- All images should include size bars.
- Images seem overexposed a bit, especially the DAPI channel. Are the authors worried about fluorescent overlap when using these images for quantification?
Reviewer 2 Report
The manuscript by Castro et al. describes the results of experiments done to determine if bacterial vaginosis associated species, Prevotella bivia and Atopobium vaginae, can incorporate into pre-established Gardnerella vaginalis biofilms. This work is driven by the hypothesis that Gardnerella spp. are responsible for initiation of biofilm formation in the transition to bacterial vaginosis and that other vaginosis associated organisms join to create multispecies biofilm. The authors also investigate if coculture of Gv with either or both of these species affects the expression of three genes, suggested to encode “virulence factors”. They conclude that both Pb and Av can incorporate into established (24 h) Gv biofilm and suggest that expression of transcripts from a Gv gene encoding a putative glycosyltransferase is higher in 3-species biofilms than when Gv is grown alone.
The questions addressed are certainly interesting and the effort to develop an experimental system for studying these challenging organisms in biofilm is worthy. Some of the results are not particularly convincing and I offer some suggestions as to how the manuscript could be improved.
Major
The stated objectives should be modified since the authors do not address “the ability of these species to interact in a multi-species biofilm”. Interaction is not addressed in the study.
In the Methods section, the authors give cell suspension concentrations (107 cfu/mL) but not volumes, which should be included. More importantly, they must describe how these concentrations were determined – colony counts on agar? quantitative PCR? OD vs. a standard curve? The numbers of bacteria added into these cultures and co-cultures is obviously a critical factor in the results so needs to be thoroughly described.
In Figure 1A, it is clear that there are differences in the amount of biofilm biomass detected at 24h for each of the species grown individually. This could be the result of different inoculum amounts (see comment above), different staining properties of the biofilms, or different growth rates. Was anything done to show that these bacteria were viable after 24 h (or 48 h) under these conditions? Is there evidence of growth? If so, this should be described. If not, it should be discussed as a factor/limitation of the study.
The observation (Figure 1B) that dual- and triple-species cultures did not result in any change in the biofilm biomass accumulated at 48 hr is noted as “curious” by the authors, which it certainly is. If the established (24 hr) Gv biofilm contains the same order of magnitude of cells as what is being added (Pb and/or Av) and those organisms are joining the biofilm and surviving/growing then there should be an increase in biomass. The results in Figure 1A and B would be more convincing if the authors presented (in the same figure), 24 hr and 48 hr biofilm biomass measurements for each combination AND each species on its own. Differences in inoculum concentration, growth rate and/or survival of each species, and limited “carrying capacity” of the environment are possible explanations that need to be discussed or addressed with experiments.
Figure 3A – If I understand the description of the experiment, Pb or Av or both were added to *established* Gv 24 h biofilms. I do not understand the statistical results shown. While it might be meaningful to compare the proportion of each non-Gv to each other (i.e. the proportion of biofilm made up of Av in Gv+Av is less/more than the proportion of Pb in Gv+Pb), the comparison to Gv is not meaningful. Also, isn’t differential growth rate a reasonable explanation for why there might be more or less of Pb or Av detected at 48 h? In other words, it’s not that Av is less able to incorporate but rather it is just slower growing.
The FISH experiments are an interesting addition, but I suggest the authors explain how they defined “incorporated”. Specifically, especially for Av with its scarcity in the biofilm, is there a possibility that the Av is simply stuck to the Gv biofilm as opposed to integrated/incorporated? These results would be much more convincing if there was an example of another bacterial species that did NOT incorporate into the Gv biofilm, thus ruling out the possibility that the Gv biofilm is simply sticky and any organism that could survive in these culture conditions could “incorporate”. It would also be beneficial to examine the planktonic fraction to see if that’s where the Av that didn’t incorporate is found as would be expected if lack of incorporation is the explanation (as opposed to lack of growth).
The inclusion of “sialidase” for the gene expression study is questionable since G. vaginalis ATCC 14018 is sialidase activity negative and lacks both of the putative extracellular sialidase enzymes found in G. piottii (Robinson et al. 2019, JBC). The gene targeted (sialidase A, sld) encodes an intracellular sialidase that is unlikely to have any role in establishing or maintaining biofilm (Kurukulasuriya et al. 2021, Infection & Immunity). This could be incorporated into the Discussion as an explanation for why no change in expression of this gene was observed.
Figure 4. The text refers to the results in terms of fold-change. The results should be expressed in the Figure on the Y-axis as fold-change as the authors have done in their previous work (ref 37).
Results regarding expression of HMPREF0424_0821 don’t match their previous study where it was more highly expressed in dual species cultures of Gv with P. bivia and A. vaginae (ref 37). There is some mention in the Discussion of the role of culture medium in affecting gene expression, but this incongruence should be specifically discussed.
Overall, I found the conclusions over-stated. The claim in the Abstract that “This study suggests that microbial interactions between co-infecting bacteria can deeply affect the progress of BV and its clinical outcome.” is not supported. The authors have provided evidence that Av and Pb can join an established Gv biofilm in vitro, and there is possibly a minor (~2-fold increase) increase in expression of one gene investigated. This borderline fold-change is not a compelling argument for biological relevance, so the conclusions should be more conservative.
Minor comments
Abstract, line 19: “incorporate INTO pre-established…”
Table 1. The table legend suggests a ranking of hybridization “quality” but there is no explanation in the methods as to how this was scored.
Figure 3 and supplemental figure – negative (unstained) controls should be included. Further to the comment above regarding Table 1, it would be ideal to include images of these negative controls to show lack of cross-hybridization.
Discussion, line 197 – “co-incubation WITH A. vaginae and P. bivia”
Line 224 – “did not have a significant effect on the enhancement in the biofilm biomass” should be “did not significantly enhance the amount of biofilm biomass” or similar
Line 231 – “particularly significant” should be “only significant” since the difference was seen only in the triple-species biofilm
Line 259 – what is the evidence that A. vaginae can survive in NYC III?
Line 266-268 – this sentence does not make sense and should be re-written for clarity
Line 269 – presumably “thinner” means lower OD in Figure 1? Since the authors did not measure biofilm thickness (just CV OD) they should correct this statement.
Line 291 – correct the strain name for A. vaginae (ATCC not ATTC)
Round 2
Reviewer 1 Report
The authors did attempt to address all issues brought up by the reviewers and improved the manuscript with clarifying text as well as the addition of some new information. This new information came in the form of microscopy and qPCR data.
- In the microscopy, addition of z stacks in the images was helpful although side stacks (images from the side of the biofilm showing the full z stack) would have been more informative. The added images don't actually show for instance thinner biofilms with Pv alone which the authors state in the discussion.
- For qPCR, addition of new genes was also helpful. There is some confusion as to the statistics. Why are there stars above the control 48 hour biofilm samples?
- Addition of clarifying information on the methods, etc. was helpful
Although the authors presented data on why their method of PNA Fish was the best method for looking at the biofilm, this was somewhat unconvincing to me in terms of still needing CFU counts. They argue that VBNC populations could affect CFU counts but VBNC are generally not an issue in populations grown in rich medium for short duration biofilms of only 48 hours. In addition, the authors used the more crude crystal violet assay rather than counts which are not descriptive of the overall population and can vary greatly depending on matrix production, etc. CFU counts would definitely have improved the ability of readers to interpret some of the data especially in light of the fact that the overall biomass in single species biofilms after 48 hours was no different from the sample samples with the addition of 10^7 CFU of two additional species (as in Figure 1). It would be very interesting to see what the makeup of the overall cell numbers are rather than only the proportion. That is not to discount PNA Fish as an important method, but to confirm and clarify some of the results.
Reviewer 2 Report
The authors have been very thorough in their response to the comments made, and have added additional information that improves the overall quality of the work. I have no further comments.
Author Response
Reviewer 2
#1. “The authors have been very thorough in their response to the comments made, and have added additional information that improves the overall quality of the work. I have no further comments.”
Author’s answer: We thank and express our appreciation to the reviewer for carefully reading the revised version of the manuscript.